# *Lactobacillus rhamnosus* Hsryfm 1301 Fermented Milk Regulates Lipid Metabolism and Inflammatory Response in High-Fat Diet Rats

Hengxian Qu [1,2], Lina Zong [1,2], Jian Sang [3], Jiaojiao Liang [1,2], Yunchao Wa [1,2], Dawei Chen [1,2], Yujun Huang [1,2], Xia Chen [1,2] and Ruixia Gu [1,2,*]

1 College of Food Science and Technology, Yangzhou University, Yangzhou 225000, China
2 Key Lab of Dairy Biotechnology and Safety Control, Yangzhou 225000, China
3 Realab Biotechnology (Beijing) Co., Ltd., Beijing 100000, China
* Correspondence: guruixia1963@163.com

**Abstract:** A rat model of disordered lipid metabolism was established to study the regulation of lipid metabolism and inflammatory response by *Lactobacillus rhamnosus* hsryfm 1301 fermented milk. The results showed that the high-fat diet caused the disorder of lipid metabolism in rats, accompanied by the occurrence of an inflammatory response. After *Lactobacillus rhamnosus* hsryfm 1301 fermented milk intervention, the blood lipid level was reduced along with the activity of aspartate aminotransferase (AST) and alanine aminotransferase (ALT) as well as triglyceride (TG) and total cholesterol (TC) contents in the liver of rats ($p < 0.05$), the fat vacuoles of rat hepatocytes were reduced, and the lipid accumulation in the rat liver was decreased. Liver injury was restored. Meanwhile, the levels of free fatty acid (FFA) and fatty acid synthase (FAS), acetyl coenzyme A carboxylase (ACC), lipoprotein esterase (LPL) and hepatic lipase (HL) in serum and liver of rats were significantly lower than those in the model group ($p < 0.05$), which indicated that fatty acid synthesis was inhibited, fatty acid production was reduced and lipid metabolism was restored to balance. In addition, the levels of reactive oxygen species (ROS) and inflammatory factors in the serum of rats were also significantly reduced ($p < 0.05$), and the inflammatory response of rats was restored. *Lactobacillus rhamnosus* hsryfm 1301 fermented milk could not only inhibit fatty acid synthase but reduce the production of excessive fatty acids, thus reducing fat accumulation, restoring the balance of lipid metabolism and alleviating the inflammatory response in rats. At the same time, it can also reduce the level of ROS through the antioxidant effect, alleviate the inflammatory response, and thus alleviate the disorder of lipid metabolism.

**Keywords:** probiotics; fermented milk; lipid metabolism; inflammatory response

## 1. Introduction

A high-fat diet has become a major cause of lipid metabolism disorders, which accelerate the onset and development of hyperlipidemia, fatty liver and obesity [1]. Lipid metabolism and inflammatory response have a reciprocal regulatory role. Lipid metabolism disorders induce inflammatory responses, while inflammatory responses promote cellular uptake and accumulation of lipids, inhibit cellular lipid efflux, and promote the development of lipid metabolism disorders. The liver is the main organ of lipid metabolism and is the site of lipid uptake, oxidation and fatty acid secretion [2]. Fatty acid metabolism in the liver plays a key role in lipid metabolism, and the intake of high-fat foods, where a large amount of fat enters the blood circulation in the form of free fatty acids and chylomicron and is then absorbed by the liver, causes disorders of liver lipid metabolism. At the same time, the increased synthesis of fatty acids by the liver using saccharides and the lipolysis of adipose tissue will continue to produce large amounts of free fatty acids [3].

Reducing the intake of high-fat foods is the main measure to alleviate lipid metabolism disorders, while good lifestyle habits are also essential [4]. In addition, drugs such as statins, clofibrates, niacin, bile acid chelators and ezetimibe are also used to treat lipid metabolism disorders. However, these drugs have some side effects on the muscle, liver or intestine [5]. Numerous studies have confirmed the role of probiotics in alleviating lipid metabolism disorders, and probiotics can lower blood lipids, prevent and alleviate hyperlipidemia and fatty liver and its complicating diseases, and restore lipid metabolism balance [6,7]. Probiotics can directly reduce absorbable cholesterol in the host intestine through co-precipitation and assimilation absorption effects and by promoting bile acid circulation [8], and can also indirectly regulate lipid metabolism by modulating the intestinal flora [9] and reducing the expression activity of host adipose storage genes [6]. In addition, the anti-inflammatory effects of probiotics have been demonstrated and the interaction between lipid metabolism and the inflammatory response has been shown, but few studies have investigated the co-regulatory effects of probiotics on lipid metabolism and inflammatory response.

A previous study found that *Lactobacillus rhamnosus* hsryfm 1301, a human-derived probiotic derived from long-lived elderly people, has strong acid and bile salt tolerance and antioxidant capacity, and has a good adjuvant hypolipidemic effect, which can exert its adjuvant hypolipidemic functional properties by regulating the lipid-related intestinal flora [9]. In this study, we used *Lactobacillus rhamnosus* hsryfm 1301 fermented milk to intervene in rats with lipid metabolism disorders. By investigating the effect of probiotics on fatty acid metabolism as well as inflammatory response in lipid metabolism, we investigated its co-regulatory effect on lipid metabolism and inflammatory response in rats.

## 2. Materials and Methods

### 2.1. Probiotic Bacteria and Fermented Milk Preparation

*Lactobacillus rhamnosus* hsryfm1301 were provided by Jiangsu Key Lab of Dairy Biological Technology and Safety Control, China. After two generations of activation, the strain was inoculated into a skim milk medium at 3% inoculum, incubated at 37 °C for 18 h, then inoculated into whole milk at 3% inoculum, and fermented at 37 °C for 72 h to prepare *Lactobacillus rhamnosus* hsryfm 1301 fermented milk.

### 2.2. Animals and Treatment

Twenty-seven healthy male Wistar rats were purchased from the Comparative Medical Center of Yangzhou University, Jiangsu, China. The rats were 6 weeks old and weighed about 218 g at the start of the experiment. The animal experiments conformed the U.S. National Institutes of Health guidelines for the care and use of laboratory animals (NIH Publication No. 85-23 Rev. 1985) and were approved by the Animal Care Committee of the Center for Disease Control and Prevention (Jiangsu, China). All animals were housed under a 12-hour light/12-hour dark cycle in a controlled room with a temperature of $23 \pm 3$ °C and a humidity of $50\% \pm 10\%$. The animals were acclimated to their new circumstances for one week. Then rats were randomly divided into 3 groups ($n = 9$): control group (C), model group (M), and *Lactobacillus rhamnosus* hsryfm1301 treatment group (P). The normal group was fed a low-fat diet (LFD: flour 20%, rice flour 10%, corn 20%, drum skin 26%, soy material 20%, fish meal 2%, bone meal 2%). Model group and *Lactobacillus rhamnosus* hsryfm1301 treatment groups were fed a high-fat diet (HFD: 10% lard, 10% egg powder, 1% cholesterol and 0.2% bile salts and 78.8% LFD) for 8 weeks. All rats were allowed free access to food and water. All rats received the following treatments by gavage: normal group and model group rats: milk (1 mL/100 g); *Lactobacillus rhamnosus* hsryfm1301 treatment group: *Lactobacillus rhamnosus* hsryfm1301 fermented milk (1 mL/100 g, $10^9$ CFU/mL). After 8 weeks, rats underwent 12 h of fasting prior to being anaesthetized and dissected. All rats were euthanized at the anestrus period following anaesthesia under 1% sodium entobarbital.

*2.3. Body Weight and Food Intake*

The body weight of rats was measured once a week, and the diet of each group was recorded daily.

*2.4. Blood Samples Handling*

Blood was obtained from each group at 8 weeks. Serum was collected by centrifugation at $3000\times g$ for 15 min and stored at $-20\,^{\circ}$C.

*2.5. Liver Samples Handling*

Part of the liver was put in cold physiologic saline immediately and tissue homogenate was prepared (10%, $w/v$). The hypothalamus homogenates were centrifuged at $4000\times g$ for 15 min at 4 $^{\circ}$C, and the supernatant was collected and stored at $-20\,^{\circ}$C until further analyzed.

*2.6. Indicator Testing*

Serum total cholesterol (TC), triglycerides (TG), high-density lipoprotein cholesterol (HDL-C), low-density lipoprotein cholesterol (LDL-C), AST and ALT, and liver TC and TG (Ningbo Meikang, Ningbo, China) were measured by a Model 7020 fully automated biochemical analyzer (Hitachi, Ibaraki Prefecture, Japan); ROS, FAS, ACC, LPL, HL, tumor necrosis factor-$\alpha$(TNF-$\alpha$), Interleukin-6(IL-6), interleukin-8(IL-8), nuclear factor kappa-B(NF-$\kappa$B), Interleukin-1$\beta$(IL-1$\beta$) and transforming growth factor-$\beta$(TGF-$\beta$) were measured by ELISA kits (Shanghai Hualan, China). FFAs were measured according to the method provided in the assay kit (Beijing Solabao, China). All samples were repeated three times.

*2.7. Experimental Pathology of Liver*

Rat liver tissues were trimmed to 0.5 cm $\times$ 0.5 cm $\times$ 0.5 cm size, fixed in 4% paraformaldehyde, and stored at 4 $^{\circ}$C. The liver placed in 4% paraformaldehyde was dehydrated and embedded and then sectioned. The sections were stained with reference to the method provided by the HE staining kit (Shanghai Biotech, Shanghai, China), observed using a microscope and photographed.

*2.8. Statistical Methods*

Statistical analysis was performed using GraphPad Prism 9 (San Diego, CA, USA). The results are presented as the means $\pm$ SEs, and the differences between the different samples were analysed using a one-way analysis of variance (ANOVA, Tukey). Values of $p < 0.05$ were considered statistically significant.

## 3. Result

*3.1. Effect of Lactobacillus rhamnosus Hsryfm 1301 Fermented Milk on the Body Weight of Rats*

Different degrees of weight gain occurred in the three groups of rats, and the weight gain in group M was significantly higher than that in group C and group P ($p < 0.05$), while there was no significant difference between group P and group C in terms of weight gain ($p > 0.05$). There was no significant difference ($p > 0.05$) in the average weekly food intake of rats in each group. *Lactobacillus rhamnosus* hsryfm 1301 fermented milk was able to slow down the weight gain of rats (Table 1).

**Table 1.** Body weight and food intake of rats (*n* = 9).

| Group | Initial Weight (g) | Final Weight (g) | Weight Gain (g) | Weekly Food Intake (g) |
|---|---|---|---|---|
| Control group (C) | 216.40 ± 6.03 [a] | 347.7 ± 13.89 [b] | 131.30 ± 11.55 [b] | 1076.13 ± 201.96 [a] |
| Model group (M) | 218.11 ± 5.15 [a] | 401.55 ± 18.76 [a] | 183.44 ± 21.18 [a] | 1086.25 ± 223.24 [a] |
| *Lactobacillus rhamnosus* hsryfm1301 treatment group (P) | 218.93 ± 6.21 [a] | 357.00 ± 18.14 [b] | 138.07 ± 19.77 [b] | 994.13 ± 161.87 [a] |

Comparison in the same column, different letters indicate significant differences (*p* < 0.05). "Weekly food intake" is the average of the total food intake of each group of rats per week.

*3.2. Effect of Lactobacillus rhamnosus Hsryfm 1301 Fermented Milk on Serum Biochemical Indexes of Rats*

Compared with rats in group C, serum TC, TG, LDL-C and FFA levels were significantly increased (Figure 1a,b,d,g) and HDL-C levels were significantly decreased (*p* < 0.05) in rats in the group M (Figure 1c), indicating that the high-fat diet caused disorders of lipid metabolism in rats. After *Lactobacillus rhamnosus* hsryfm 1301 fermented milk intervention, serum TC, TG, LDL-C and FFA levels were significantly decreased (*p* < 0.05) and HDL-C levels were significantly increased (*p* < 0.05). All these indicate that *Lactobacillus rhamnosus* hsryfm 1301 fermented milk has the functional property of adjuvant hypolipidemic in vivo and can improve the blood lipid level and the FFA in the serum of rats was reduced.

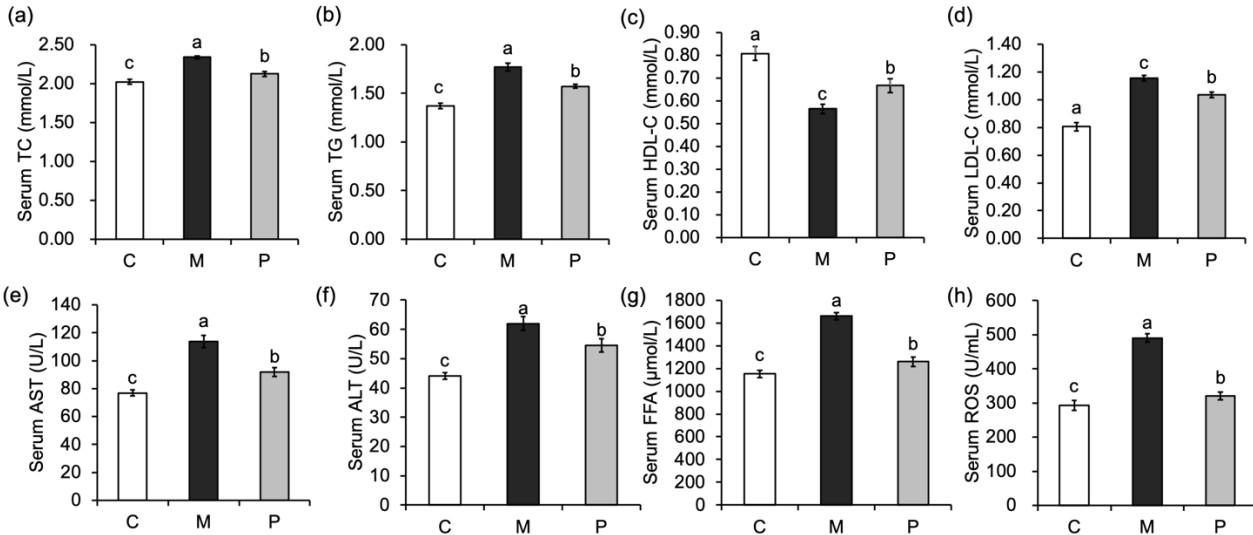

**Figure 1.** Effect of *Lactobacillus rhamnosus* hsryfm 1301 fermented milk on serum biochemical indices of rats (*n* = 9). (**a**) Rats serum TC, (**b**) rats serum TG, (**c**) rats serum HDL-C, (**d**) rats serum LDL-C, (**e**) rats serum AST, (**f**) rats serum ALT, (**g**) rats serum FFA, (**h**) rats serum ROS. Different letters indicate significant differences (*p* < 0.05), same below.

Compared with group C, the serum levels of ALT, AST and ROS were significantly increased in group M (*p* < 0.05) (Figure 1e,f), indicating that the high-fat diet caused liver damage in rats. After the intervention of *Lactobacillus rhamnosus* hsryfm 1301 fermented milk, ALT, AST and ROS were improved to different degrees, and all were significantly decreased (*p* < 0.05), and *Lactobacillus rhamnosus* hsryfm 1301 fermented milk could effectively alleviate the liver injury caused by a high-fat diet.

### 3.3. Effect of Lactobacillus rhamnosus Hsryfm 1301 Fermented Milk on Liver Injury of Rats

The liver FFA, TC and TG levels were significantly increased in group M compared with group C ($p < 0.05$). After the intervention of *Lactobacillus rhamnosus* hsryfm 1301 fermented milk, the liver FFA, TC and TG contents of rats in the group P were significantly reduced compared with those in the group M ($p < 0.05$) (Figure 2a–c), indicating that *Lactobacillus rhamnosus* hsryfm 1301 fermented milk could reduce the fat accumulation and free fatty acids in liver tissues, and had a modulating effect on the disorder of lipid metabolism induced by a high-fat diet.

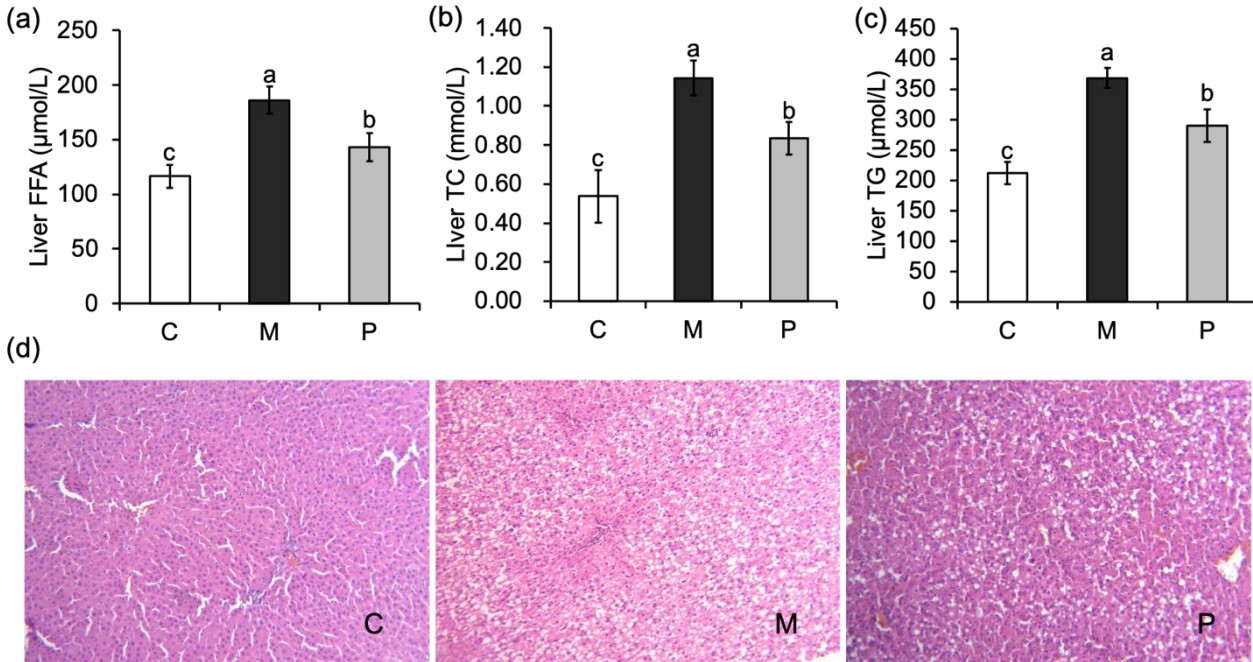

**Figure 2.** Effect of *Lactobacillus rhamnosus* hsryfm 1301 fermented milk on the liver injury of rats ($n = 9$). (**a**) Rats liver FFA, (**b**) rats liver TC, (**c**) rats liver TG, (**d**) histopathological observation of liver (×100).

The pathological sections of rat liver showed that the hepatic lobules of rats in group C were structurally intact, with clear cell boundaries and no fat vacuoles were seen. Rats in group M had severely damaged hepatocyte structure, unclear cell gaps and obvious fat deposition, and a large number of fat vacuoles of hepatocytes were seen. After the intervention of *Lactobacillus rhamnosus* hsryfm 1301 fermented milk, compared with the group M, the fat vacuoles of liver cells were reduced (Figure 2d), and the liver injury was alleviated, further confirming the regulation of lipid metabolism disorder by *Lactobacillus rhamnosus* hsryfm 1301 fermented milk.

### 3.4. Effect of Lactobacillus rhamnosus Hsryfm 1301 Fermented Milk on Liver Fatty Acid Metabolism of Rats

The levels of liver FAS and ACC were significantly increased in group M compared to group C ($p < 0.05$), indicating that a high-fat diet led to increased liver de novo synthesis of fatty acid in rats (Figure 3a,b). After *Lactobacillus rhamnosus* hsryfm 1301 fermented milk intervention, the levels of liver FAS and ACC in rats were significantly reduced compared with group M ($p < 0.05$), suggesting that *Lactobacillus rhamnosus* hsryfm 1301 fermented milk was able to reduce fatty acid synthesis and inhibit fat accumulation in the liver.

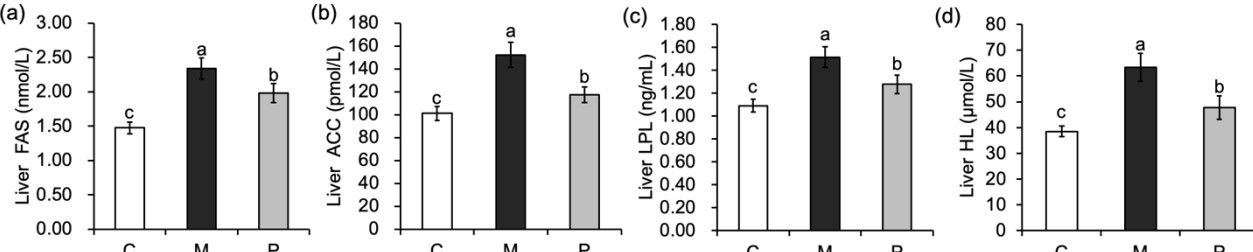

**Figure 3.** Effect of *Lactobacillus rhamnosus* hsryfm 1301 fermented milk on liver fatty acid metabolism of rats (*n* = 9). (**a**) Rats liver FAS, (**b**) rats liver ACC, (**c**) rats liver LPL, (**d**) rats liver HL.

The levels of liver LPL and HL were significantly increased ($p < 0.05$) in group M compared with group C (Figure 3c,d), indicating that the rate of lipolysis in the liver of rats was exacerbated by the intake of a high-fat diet. After *Lactobacillus rhamnosus* hsryfm 1301 fermented milk intervention, the levels of liver LPL and HL in rats in the group P were significantly decreased ($p < 0.05$) compared with the group M, and fat lipolysis gradually returned to the normal rate.

*3.5. Effect of Lactobacillus rhamnosus Hsryfm 1301 Fermented Milk on Serum Inflammatory Factors in Rats*

The levels of serum inflammatory factors TNF-α, IL-6, IL-8, NF-κB, IL-1β and TGF-β1 were significantly increased ($p < 0.05$) in group M compared with group C (Figure 4), and the high-fat diet caused the development of chronic inflammatory responses in rats. After *Lactobacillus rhamnosus* hsryfm 1301 fermented milk intervention, compared with group M, serum inflammatory factors were significantly reduced ($p < 0.05$), in which TNF-α, IL-1β and TGF-β1 were restored to levels similar to those in group C. *Lactobacillus rhamnosus* hsryfm 1301 fermented milk was able to regulate the inflammatory response caused by disorders of lipid metabolism.

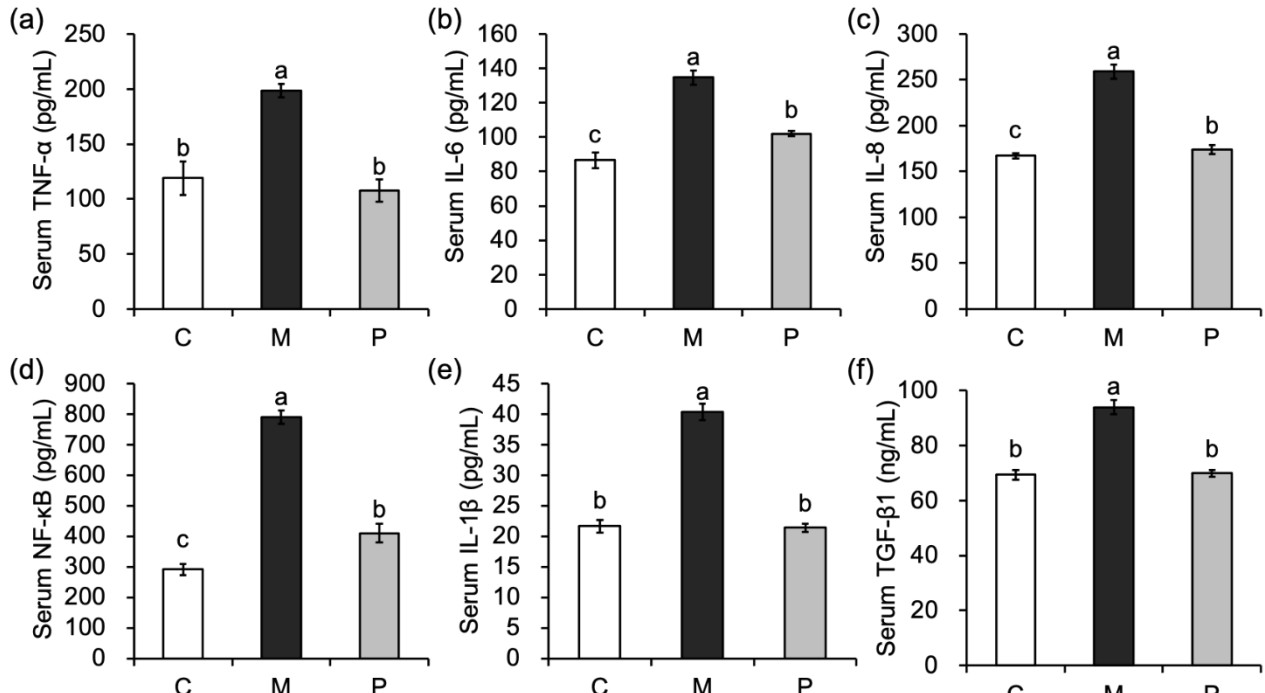

**Figure 4.** Effect of *Lactobacillus rhamnosus* hsryfm 1301 fermented milk on serum inflammatory response of rats (*n* = 9). (**a**) Rats serum TNF-α, (**b**) rats serum IL-6, (**c**) rats serum IL-8, (**d**) rats serum NF-κB, (**e**) rats serum IL-1β, (**f**) rats serum TGF-β1.

## 4. Discussion

Continuous feeding of a high-fat diet increased body weight, dyslipidemia and accumulation of free fatty acids in rats and caused fat accumulation in the liver, causing disorders of lipid metabolism and accompanying inflammatory reactions. However, continuous gavage of *Lactobacillus rhamnosus* hsryfm 1301 fermented milk while on a high-fat diet significantly reduced body weight, regulated the recovery of blood lipid levels, inhibited liver fatty acid synthesis enzymes FAS and ACC, reduced free fatty acids in serum and liver, and alleviated inflammatory reactions in rats. It suggested the co-regulatory effect of *Lactobacillus rhamnosus* hsryfm 1301 fermented milk on lipid metabolism disorder and inflammatory response.

Previous studies in our laboratory have identified the adjuvant hypolipidemic function of *Lactobacillus rhamnosus* hsryfm 1301. *Lactobacillus rhamnosus* hsryfm 1301 was able to reduce cholesterol absorption by co-precipitation and other means [9]. In the present study, *Lactobacillus rhamnosus* hsryfm 1301 fermented milk significantly reduced serum TC, TG and LDL-C ($p < 0.05$) and significantly increased HDL-C ($p < 0.05$) while serum inflammatory factors were also significantly reduced ($p < 0.05$). Cholesterol itself is an inflammatory factor, and when there is too much of it, it leads to an inflammatory environment [10]. It has been shown that LDL and its modifiers are the main inducers of inflammation. Phospholipid-hydrolyzed LDL (LDL-X) is an activator of the p38 pathway and induces downstream IL-6 and IL-8 expression [11]. Oxidized low-density lipoprotein (ox-LDL) is another lipoprotein with pro-inflammatory effects that increases the synthesis and secretion of IL-1β while inducing NF-κB activation [12]. HDL, on the other hand, has a powerful anti-inflammatory effect, blocking the secretion of inflammatory cytokines TNF-α, IL-6, and IL-8 [13,14]. These suggested that *Lactobacillus rhamnosus* hsryfm 1301 is able to restore the balance of lipid metabolism and alleviate the inflammatory response through its hypocholesterolemic effect.

In this study, it was also found that the levels of fatty acids in both serum and liver were significantly reduced ($p < 0.05$) after *Lactobacillus rhamnosus* hsryfm 1301 fermented milk intervention, as well as the key enzymes for fatty acid synthesis, FAS and ACC, and the key enzymes for lipolysis, LPL and HL ($p < 0.05$). FAS and ACC are the key enzymes for fatty acid production during the ab initio synthesis of fatty acids enzymes [15,16]. In a healthy liver, de novo synthesis of fatty acid is not a major source of hepatic fatty acids. However, in the case of dysfunctional lipid metabolism, the liver can produce more than 25% of hepatic adipose reserves through de novo synthesis of fatty acid [17]. The reduction of FAS and ACC indicates that *Lactobacillus rhamnosus* hsryfm 1301 fermented milk can reduce fatty acid synthesis by inhibiting the key enzymes of de novo synthesis of fatty acid and reduce the level of FFA to restore the balance of dysfunctional lipid metabolism. LPL and HL are key enzymes in lipolysis [18,19] and are important lipases for the breakdown of TG. The reduction of LPL and HL was a side effect of the restoration of lipid metabolic balance, which in turn allowed lipolysis to return to normal rates.

The overproduction of fatty acids due to a high-fat diet also induces the production of ROS [20], which not only causes lipid oxidative damage in the liver but also leads to the development of inflammatory responses. In the present study, *Lactobacillus rhamnosus* hsryfm 1301 fermented milk intervention significantly reduced the level of ROS, and case-based observations revealed that hepatic lipid oxidative damage was alleviated, as well as the inflammatory response. This was attributed, on the one hand, to the reduction of FFA. On the other hand, our previous studies found that *Lactobacillus rhamnosus* hsryfm 1301 fermented milk has a strong antioxidant effect and can enhance liver antioxidant enzymes to scavenge free radicals and reduce lipid peroxide production. Thus, we hypothesized that *Lactobacillus rhamnosus* hsryfm 1301 fermented milk could also reduce ROS and alleviate inflammatory response through its antioxidant effect. In addition, previous studies have also found that *Lactobacillus rhamnosus* hsryfm 1301 is able to regulate intestinal flora and reduce the abundance of harmful bacteria in the intestine [9], which may account for its anti-inflammatory effects. *Lactobacillus rhamnosus* hsryfm 1301 fermented milk is also able to alleviate lipid metabolism disorders to some extent through its anti-inflammatory effect.

In conclusion, *Lactobacillus rhamnosus* hsryfm 1301 fermented milk has a regulating effect on lipid metabolism and inflammatory response in high-fat diet rats. *Lactobacillus rhamnosus* hsryfm 1301 fermented milk restores the balance of lipid metabolism by regulating blood lipid and fatty acid metabolism, which in turn alleviates the inflammatory response. At the same time, *Lactobacillus rhamnosus* hsryfm 1301 fermented milk can also reduce the level of ROS through an antioxidant effect and alleviate the inflammatory response, which in turn inhibits the uptake and accumulation of lipids by cells and alleviates the disorder of lipid metabolism.

**Author Contributions:** Conceptualization, H.Q., L.Z. and J.S.; methodology, H.Q. and J.L.; software, H.Q.; validation, H.Q. and Y.W.; formal analysis, H.Q.; investigation, D.C.; resources, Y.H.; data curation, H.Q.; writing—original draft preparation, H.Q.; writing—review and editing, H.Q. and R.G.; visualization, H.Q.; supervision, R.G.; project administration, X.C.; funding acquisition, R.G. All authors have read and agreed to the published version of the manuscript.

**Funding:** This research was funded by Student's Platform for Innovation and Entrepreneurship Training Program of Jiangsu grant number KYCX20-2981 and funded by the National Natural Science Foundation of China grant number 32272362.

**Institutional Review Board Statement:** The study was conducted in accordance with the U.S. National Institutes of Health guidelines for the care and use of laboratory animals (NIH Publication No. 85-23 Rev. 1985), and approved by the Animal Care Committee of the Center for Disease Control and Prevention (Jiangsu, China) (No. 202103262, 5 March 2021).

**Informed Consent Statement:** Not applicable.

**Data Availability Statement:** The data supporting the findings reported here are available on reasonable request from the corresponding author.

**Conflicts of Interest:** The authors declare that they have no conflict of interest.

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
