# Peer review of "Lactobacillus rhamnosus Hsryfm 1301 Fermented Milk Regulates Lipid Metabolism and Inflammatory Response in High-Fat Diet Rats"

_fermentation, doi:10.3390/fermentation8110584_

Round 1

Reviewer 1 Report

This work investigates the effect of a fermented milk of L. rhamnosus hsryfm 1301 over lipid metabolism regulation and inflammatory response in a hyperlipidaemia model of rats. This work demonstrates the positive effect of this probiotic over cholesterol parameters, together with an anti-inflammatory response. The results are interesting, novel and well presented. However, some aspects that might be clarified.

 §  First, authors do not mention the diet intake along the experiment. Did animals from the groups M and P differ in their energy intake? If this is the case, this might have partially affected to the results. This aspect must be clarified.

§  Figure 1: The axis of the different graphs should start at zero. Some of the differences appear exaggerated, and the statistics already explain whether these effects are sufficiently evident.

§  Do authors have any hypothesis about the L. rhamnosus mechanism of action for the lipid-reducing effect?

§  What would be the equivalent dose in humans for this probiotic? It could be applicable?

§  Minor change: There is a mistake where authors mentioned “All rats received the following treatments by lavage (gavage):”

Author Response

Dear reviewer:

Thank you for reviewing the manuscript and giving us your valuable comments. The answers to your questions are as follows.

Q:  §  First, authors do not mention the diet intake along the experiment. Did animals from the groups M and P differ in their energy intake? If this is the case, this might have partially affected to the results. This aspect must be clarified.

A: Food intake had been supplemented in Table 1, with no significant difference between the group M and P.

Q: §  Figure 1: The axis of the different graphs should start at zero. Some of the differences appear exaggerated, and the statistics already explain whether these effects are sufficiently evident.

A: The axis of the different graphs in figure1 had been changed to start at zero.

Q: §  Do authors have any hypothesis about the L. rhamnosus mechanism of action for the lipid-reducing effect?

A: The function of probiotics is strain-specific, so this paper focuses only on the mechanism of Lactobacillus rhamnosus hsryfm1301. However, in combination with the available studies, we suggest that co-precipitation and assimilation and absorption effects are key for Lactobacillus rhamnosus to effect lipids. In addition, it may indirectly regulate lipid metabolism through modulation of intestinal flora, anti-inflammatory, antioxidant and reduction of host lipid storage gene expression activity.

Q: §  What would be the equivalent dose in humans for this probiotic? It could be applicable?

A: In probiotic fermented milk, the active probiotic is an important basis for the functional effect. In this study, the viable count of Lactobacillus rhamnosus hsryfm 1301 fermented milk was 109 CFU/ml, and the daily dose for rats was 109 CFU/100 g body weight. In the pharmacological experiment, the equivalent dose for rats is approximately equivalent to 6.3 times that of human, based on the dose per unit body weight. So the dose converted to human is 1.58×109 CFU/kg body weight. This means that a 70kg adult would need to intake 1.1×1011CFU of live bacteria. One cup of 150-200ml of Lactobacillus rhamnosus hsryfm 1301 fermented milk can meet this dose requirement.

Q: §  Minor change: There is a mistake where authors mentioned “All rats received the following treatments by lavage (gavage):”

A: This error had been corrected.

Thank you and best regards.

Yours sincerely,

Hengxian Qu

Reviewer 2 Report

This study used Lactobacillus rhamnosus hsryfm 1301 fermented milk to intervene in rats with lipid metabolism disorders. By investigating the effect of probiotics on fatty acid metabolism as well as inflammatory response in lipid metabolism, the authors investigated its co-regulatory effect on lipid metabolism and inflammatory response in rats. It is well done, presented and discussed work

I just have few minor corrections

Delete (in rat( at line 3 of abstract

Describe the basis of using Lb rahmnosus in this study

Table 1 please show all the group treatments in more details

Carefully check and correct microbial formatting at references 7,9, and others

Confirm that all journal names should be start with Capital letter see for example Ref. 4,8,19 check all

Author Response

Dear reviewer:

Thank you for reviewing the manuscript and giving us your valuable comments. The answers to your questions are as follows.

Q: Delete (in rat( at line 3 of abstract

A: This error had been corrected.

Q: Describe the basis of using Lb rahmnosus in this study

A: In the third paragraph of the introduction, we had briefly described the reasons for using Lactobacillus rhamnosus hsryfm1301. It was mainly due to the safety of its source and its good basal properties. We wanted to further refine the mechanism of Lactobacillus rhamnosus hsryfm1301 to regulate lipid metabolism. It also provided support to refine the mechanism of action of Lactobacillus rhamnosus on lipid metabolism disorders.

Q: Table 1 please show all the group treatments in more details

A: The details of each group treatments had been added in table.

Q: Carefully check and correct microbial formatting at references 7,9, and others

A: The microbial formatting at references had been checked and corrected.

Q: Confirm that all journal names should be start with Capital letter see for example Ref. 4,8,19 check all

A: All journal names had been checked and corrected.

Thank you and best regards.

Yours sincerely,

Hengxian Qu